**Investigation**

# Using encrypted genotypes and phenotypes for collaborative genomic analyses to maintain data confidentiality

Tianjing Zhao (ID) ,[1,2] Fangyi Wang,[3] Richard Mott (ID) ,[4] Jack Dekkers (ID) ,[5] Hao Cheng (ID) [1,*]

[1]Department of Animal Science, University of California, Davis, CA 95616, USA
[2]Department of Animal Science, University of Nebraska-Lincoln, Lincoln, NE 68583, USA
[3]Department of Plant Sciences, University of California, Davis, CA 95616, USA
[4]Genetics Institute, University College London, London, WC1E 6BT, UK
[5]Department of Animal Science, Iowa State University, Ames, IA 50011, USA

*Corresponding author: Department of Animal Science, University of California, Davis, CA 95616, USA. Email: qtlcheng@ucdavis.edu.

To adhere to and capitalize on the benefits of the FAIR (findable, accessible, interoperable, and reusable) principles in agricultural genome-to-phenome studies, it is crucial to address privacy and intellectual property issues that prevent sharing and reuse of data in research and industry. Direct sharing of genotype and phenotype data is often prohibited due to intellectual property and privacy concerns. Thus, there is a pressing need for encryption methods that obscure confidential aspects of the data, without affecting the outcomes of certain statistical analyses. A homomorphic encryption method for genotypes and phenotypes (HEGP) has been proposed for single-marker regression in genome-wide association studies (GWAS) using linear mixed models with Gaussian errors. This methodology permits frequentist likelihood-based parameter estimation and inference. In this paper, we extend HEGP to broader applications in genome-to-phenome analyses. We show that HEGP is suited to commonly used linear mixed models for genetic analyses of quantitative traits including genomic best linear unbiased prediction (GBLUP) and ridge-regression best linear unbiased prediction (RR-BLUP), as well as Bayesian variable selection methods (e.g. those in Bayesian Alphabet), for genetic parameter estimation, genomic prediction, and GWAS. By advancing the capabilities of HEGP, we offer researchers and industry professionals a secure and efficient approach for collaborative genomic analyses while preserving data confidentiality.

Keywords: homomorphic encryption; mixed model; genomic prediction; GWAS; joint analysis

## Introduction

To conform to and capitalize on the benefits of the FAIR (findable, accessible, interoperable, and reusable) principles in agricultural genome-to-phenome studies, it is necessary to address privacy and intellectual property issues that may prevent sharing and reuse of data in research and industry. First, sharing and reuse of genotypic and phenotypic data enables reproducible research, where researchers can confirm published analyses with minimal effort. Second, for traits that are hard or expensive to measure, a single research group may have limited data for genetic analysis, which may lead to less reliable and underpowered results. This problem may be alleviated by joint analyses that include data from multiple contributors.

Although data sharing and reuse will bring significant benefits to genome-to-phenome studies in both academia and industry, it is often prohibitive to directly share raw genotype and phenotype data due to privacy concerns, commercial interests, and data-sharing policies, and because the risks of sharing raw data may not be fully understood by the data owners. For example, although individual identifiers can be anonymized, information about an anonymized individual might still be disclosed by comparing its genotypes to known genotyped relatives. To avoid the concerns about sharing raw data, consortia are often established, and raw data are only shared with members of the consortium or with researchers who are approved for access. In other cases, external researchers may perform analysis on the data owner's computer system without access to the raw data. These approaches, however, still pose risks to privacy and intellectual property, hampering widespread data sharing and reuse.

Homomorphic encryption (HE) refers to a type of encryption of raw data (hereafter referred to as "plaintext") in a manner that obscures confidential aspects of the data, while certain computations on the encrypted data (hereafter referred to as "cyphertext") match the results from the plaintext, when decrypted. While several methods for HE have been proposed for genomic analysis, most limit the types of computations and analyses that can be conducted on the encrypted data (cyphertext). For example, for case–control genome-wide association studies (GWAS), HE schemes were proposed to calculate allelic chi-square test and perform logistic regression (Lu *et al.* 2015; Chen *et al.* 2018; Blatt *et al.* 2020; Jie Sim *et al.* 2020). However, these methods ignore random and fixed effects that account for family and population admixture. Although linear mixed models are widely used in genetic

analyses such as genomic prediction and GWAS (Bradbury *et al.* 2007), the use of HE for mixed models is scarce.

Recently, Mott *et al.* (2020b) proposed an encryption method, called homomorphic encryption for genotypes and phenotypes (HEGP), that is specifically suited to single-marker regression in GWAS using linear mixed models with Gaussian errors. HEGP is based on high-dimensional random orthogonal transformations of the plaintext that encrypts phenotypes, genotypes, and specified covariates by replacing them with random linear superpositions, such that cyphertext genotypes and phenotypes cannot be linked back to individual identifiers. HEGP preserves linkage disequilibrium between markers but scrambles the genomic relationship between individuals. Moreover, under a linear mixed model with Gaussian errors, the likelihood of the cyphertext is unchanged, such that the encryption does not affect the outcomes of single-marker regression in GWAS analyses. HEGP differs conceptually from other HE methods in that some outputs (particularly the parameter estimates) are unaffected by encryption and do not need to be decrypted.

In this paper, we extend the HEGP scheme for wider application in genome-to-phenome analyses. We demonstrate that HEGP can be effectively applied to many popular mixed models, beyond single-marker regression. These models, including Bayesian variable selection methods such as those in Bayesian Alphabet, are routinely employed in the fields of animal and crop improvement, for genetic analyses of quantitative traits, including genetic parameter estimation, genomic prediction, and GWAS. We show how most of the quantitative genetics toolbox used by animal and plant breeders can be integrated with data-sharing protocols and performed while protecting important types of potentially confidential or commercially sensitive information.

## Materials and methods
### HE using high-dimensional random orthogonal matrix

We will use **y**, a vector of length $n$, to denote the plaintext phenotypes for $n$ observations, and **M** to denote the $n \times p$ plaintext genotype covariate matrix for the $n$ observations across $p$ single nucleotide polymorphism (SNPs). To infer unknowns in mixed models, these quantities will typically be used in the multiplicative forms $\mathbf{M}^T\mathbf{M}$ and $\mathbf{M}^T\mathbf{y}$. Thus, intuitively, any data encryption scheme that leaves the above multiplications unchanged would produce the same GWAS and genomic prediction outcomes.

HEGP uses a high-dimensional random $n \times n$ orthogonal matrix **P**, such that $\mathbf{P}^T\mathbf{P} = \mathbf{I}$ and the determinant $|\mathbf{P}| = 1$. The suitable choices of **P** for the purpose of encryption are discussed in a later section. The plaintext genotypes and phenotypes are encrypted as

$$\mathbf{M}^* = \mathbf{PM},$$
$$\mathbf{y}^* = \mathbf{Py}, \tag{1}$$

because

$$\mathbf{M}^T\mathbf{M} = \mathbf{M}^T\mathbf{P}^T\mathbf{PM} = (\mathbf{PM})^T(\mathbf{PM}) = \mathbf{M}^{*T}\mathbf{M}^* \tag{2}$$

and

$$\mathbf{M}^T\mathbf{y} = \mathbf{M}^T\mathbf{P}^T\mathbf{Py} = (\mathbf{PM})^T\mathbf{y}^* = \mathbf{M}^*\mathbf{y}^*. \tag{3}$$

In contrast to other methods of HE, the outputs of HEGP (i.e. marker effect estimates and *P*-values) are automatically plaintext,

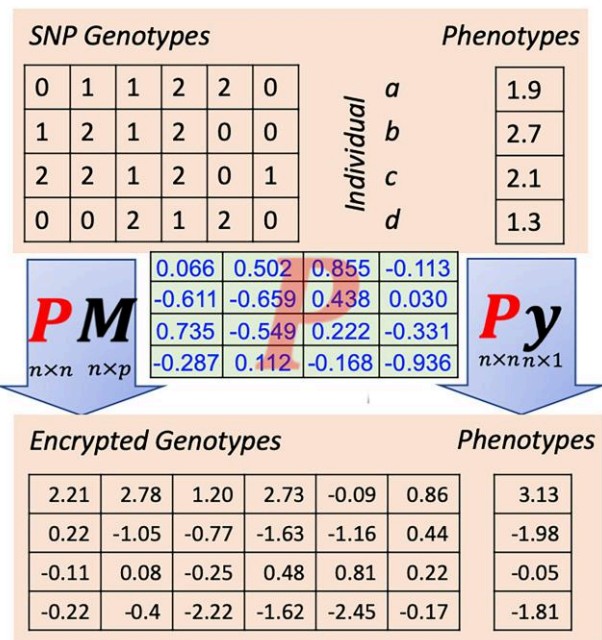

**Fig. 1.** Illustration of HEGP. For 4 individuals (a–d), the raw genotypes for 6 SNPs and the raw phenotypes are provided in the upper part of the figure. The raw data are encrypted by premultiplying with a random orthogonal encryption matrix **P**, which is displayed in the middle of the figure. The encrypted phenotypes and genotypes, shown in the lower part of the figure, are "random" linear combinations of the raw phenotypes and genotypes of the original four individuals (a–d).

regardless of whether the inputs are plaintext or cyphertext. This means that there is no need to decrypt the outputs such as marker effect estimates and hence no decryption key is distributed. In a later section, we will show that the HEGP does not affect the inference of marker effects, thus with the plaintext of genotypes, the estimated breeding values (EBV) can be calculated. Otherwise, the EBV calculated from the cybertext of genotypes remains cybertext EBV.

### *Conceptual overview*
Figure 1 illustrates HEGP for a small example of 4 individuals (a–d) and 6 SNPs. By multiplying by the random orthogonal matrix **P**, phenotypes and genotypes in the encrypted data become "random" linear combinations of phenotypes and genotypes of the original 4 individuals (a–d).

Figure 2 compares plaintext and cyphertext genotypes from a larger pig dataset (Cleveland *et al.* 2012). Figure 2a and b presents the heat maps of the plaintext and cyphertext genotypes. For the plaintext, each row represents an individual and each column represents the genotypes for an SNP across individuals. As shown in Fig. 2c and d, after encryption, the genotypes transform from trimodal values (0/1/2) to continuous values that closely resemble a sample from a normal distribution.

The set of orthogonal $n \times n$ matrices forms a group under multiplication and includes the identity matrix, which is clearly ineffective for encryption. Therefore, it is necessary for **P** to be randomly generated and independent of the plaintext. A suitable method is derived from the Stiefel manifold, or Haar measure (Chikuse and Chikuse 2003; Hoff 2009), which is measure-preserving, meaning that the measure (loosely speaking, the sampling probability) of any data matrix **M** is the same as the measure of **PM**. For this method, first an $n \times n$ matrix **B** is generated, whose

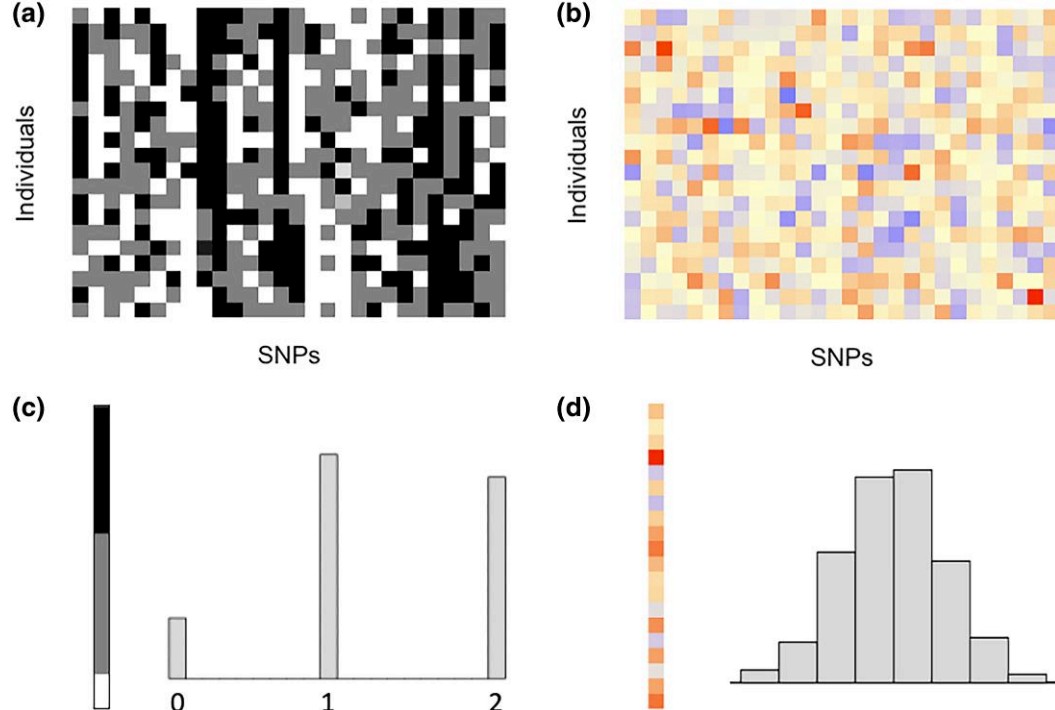

**Fig. 2.** a) A subset of pig genotypes provided in the dataset of Cleveland *et al.* (2012). Genotypes are coded as 0, 1, and 2, which are presented by white, gray, and black colors, respectively. Each row represents one individual, and each column represents one marker. b) The corresponding encrypted genotypes, encrypted via a high-dimensional random orthogonal matrix. The encrypted genotypes are a continuum of real numbers presented by different colors. c) Sorted genotypes of one marker, coded as 0/1/2 (left), and its trimodal distribution (right). d) The corresponding encrypted genotypes [same order as in c)] and their bell-shaped distribution.

entries are sampled independently from a standard normal distribution. Next, an $n \times n$ random orthogonal matrix is generated as $\mathbf{P} = \mathbf{B}(\mathbf{B}^T\mathbf{B})^{-\frac{1}{2}}$. In detail, $(\mathbf{B}^T\mathbf{B})^{-\frac{1}{2}}$ is computed as $\mathbf{Q}\mathbf{\Lambda}^{-\frac{1}{2}}\mathbf{Q}^T$, where $\mathbf{Q}$ and $\mathbf{\Lambda}$ are obtained from the eigen decomposition of $\mathbf{B}^T\mathbf{B}$, i.e. $\mathbf{B}^T\mathbf{B} = \mathbf{Q}\mathbf{\Lambda}\mathbf{Q}^T$. Matrix $\mathbf{P}$ is easily seen to be orthogonal (by checking that $\mathbf{P}^T\mathbf{P} = \mathbf{I}$ ) and furthermore can be shown to be randomly sampled from the Stiefel manifold. The R package rstiefel (Hoff 2012) can be used to generate $\mathbf{P}$.

### Relationships between SNPs and between individuals

HEGP preserves relationships between genotypes (i.e. linkage disequilibrium, $r^2$), but scrambles relationships between individuals. Any orthogonal transformation preserves the dot product of two vectors and, geometrically, acts as a rotation of a hyper-sphere in which SNP genotype vectors and phenotype vectors are represented as points on its surface. The cosine of the angle between any pair of points subtended at the origin equals their Pearson correlation coefficient, or dot product and a rotation merely changes the coordinate system while leaving angles unchanged. In HEGP, all marker genotypes (and phenotypes) are rotated by the same orthogonal matrix as $\mathbf{PM} = [\mathbf{Pm}_1, \ldots, \mathbf{Pm}_p]$. Thus, the LD between $j$th and $k$th marker is preserved since

$$\begin{aligned}(\mathbf{m}_j^*)^T(\mathbf{m}_k^*) &= (\mathbf{Pm}_j)^T(\mathbf{Pm}_k) \\ &= \mathbf{m}_j^T\mathbf{m}_k,\end{aligned} \quad (4)$$

where we use the fact that for any orthogonal matrix, $\mathbf{P}^T\mathbf{P} = \mathbf{I}$. For illustration, the LD matrices based on the raw and the encrypted genotypes for 5,000 markers in the pig dataset of Cleveland *et al.* (2012) are shown in Fig. 3a. The LD matrix is calculated as $\frac{1}{n}\mathbf{H}^T\mathbf{H}$,

where $\mathbf{H}$ is the normalized genotype matrix. The $j$th marker of the $i$th individual is normalized as $H_{i,j} = \frac{M_{i,j} - 2p_j}{\sqrt{2p_j(1-p_j)}}$, where $p_j$ is the allele frequency. In Fig. 3a, the 2 LD matrices are almost identical, and the correlation between elements in the 2 LD matrices is 1.0.

In contrast to LD relationships, HEGP scrambles relationships between individuals since $(\mathbf{PM})(\mathbf{PM})^T = \mathbf{P}(\mathbf{MM}^T)\mathbf{P}^T$ and, after transformation, individual records are random linear combinations of the original records. For demonstration, GRMs, calculated as $\frac{1}{p}\mathbf{HH}^T$, based on plaintext and cyphertext genotypes, are shown in Fig. 3b for a subset of the pig dataset. The elementwise correlation between the 2 GRM is $\sim 0$.

## Statistical preliminaries

As we will demonstrate, in addition to single-marker regression for GWAS using linear mixed models with Gaussian errors, HEGP is compatible with most genetic analyses that use mixed models, including GBLUP, SNP-BLUP, Bayesian Alphabet, and others.

### Mixed models

The mixed model is a cornerstone for many quantitative genetic analyses, including genetic parameter estimation, genomic prediction, and GWAS (Meuwissen *et al.* 2001; VanRaden 2008; Hayes *et al.* 2009; Wang *et al.* 2012; Moser *et al.* 2015a; Wang *et al.* 2016; Fernando *et al.* 2017; Legarra *et al.* 2018). In particular, GBLUP (Habier *et al.* 2007; VanRaden 2008; Hayes *et al.* 2009) is one of the most widely used linear mixed models for genomic prediction. The GBLUP model can be written as

$$\mathbf{y} = \mathbf{X}\boldsymbol{\beta} + \mathbf{u} + \mathbf{e}, \quad (5)$$

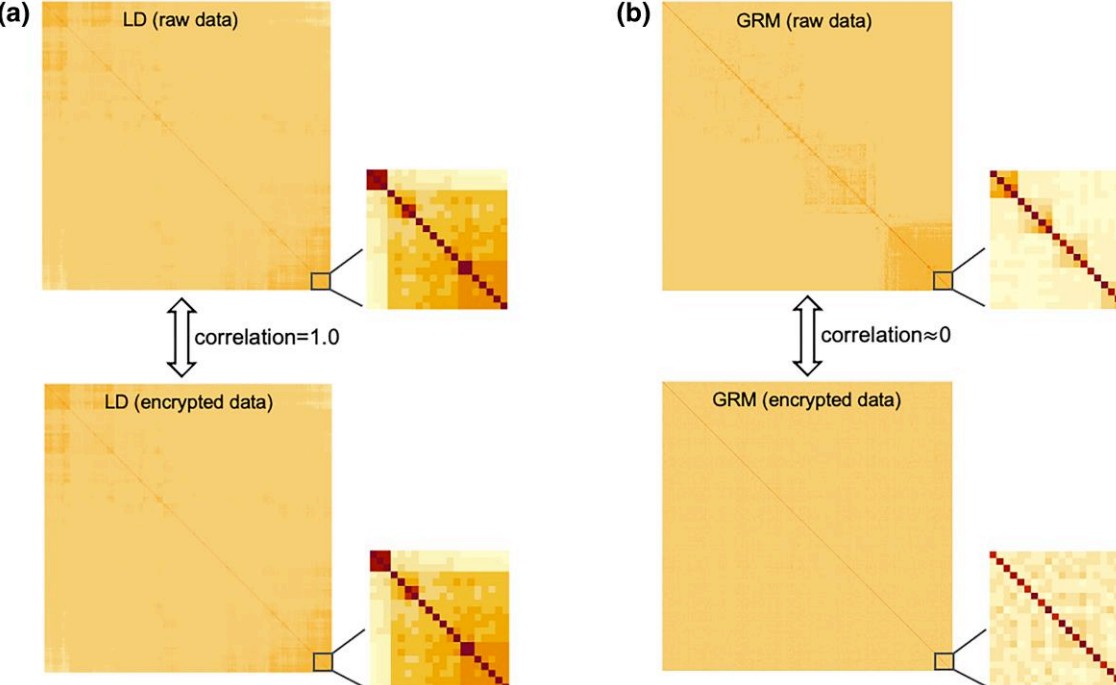

**Fig. 3.** a) Linkage disequilibrium (LD) matrix calculated using raw genotypes (up), and encrypted genotypes (down). The LD matrix is preserved using encrypted data, and the correlation between the 2 LD matrices is 1.0. b) GRM calculated using the raw genotypes (up), and encrypted genotypes (down). The GRM is scrambled using encrypted data, and the correlation between 2 GRMs is close to 0.

where $\mathbf{y}$ is a vector of phenotypes of length $n$, and $\mathbf{X}$ is the incidence matrix for nongenetic fixed effects, denoted by $\boldsymbol{\beta}$. Vector $\mathbf{u}$ contains additive genetic values for $n$ individuals and follows a multivariate normal distribution $\mathbf{u} \sim N(\mathbf{0}, \mathbf{G}\sigma_u^2)$, where $\mathbf{G}$ is the GRM proportional to $\mathbf{MM}^T$, where $\mathbf{M}$ is an $n \times p$ genotype covariate matrix, and $\sigma_u^2$ is the genetic variance. Vector $\mathbf{e}$ includes $n$ random residuals and follows a normal distribution, $\mathbf{e} \sim N(\mathbf{0}, \mathbf{I}\sigma_e^2)$, where $\sigma_e^2$ is the residual variance. Narrow sense heritability is defined to be $h^2 = \frac{\sigma_u^2}{(\sigma_u^2 + \sigma_e^2)}$.

The GBLUP model is equivalent to the following marker effects model (hereinafter referred to as SNP-BLUP) in terms of predicting genetic values (Fernando 1998; Habier *et al.* 2007; Strandén and Garrick 2009):

$$\mathbf{y} = \mathbf{X}\boldsymbol{\beta} + \mathbf{M}\boldsymbol{\alpha} + \mathbf{e}, \tag{6}$$

where $\boldsymbol{\alpha}$ is a vector of $p$ additive marker effects, with $\boldsymbol{\alpha} \sim N(\mathbf{0}, \mathbf{I}\sigma_\alpha^2)$. The same point estimates of marker effects $\hat{\boldsymbol{\alpha}}$ can be obtained from the estimated genetic values $\hat{\mathbf{u}}$ in GBLUP as $\hat{\boldsymbol{\alpha}} = \mathbf{M}^T(\mathbf{MM}^T)^{-1}\hat{\mathbf{u}}$.

The Gaussian prior distribution of marker effects in SNP-BLUP is just one, analytically tractable, member of the "Bayesian Alphabet", in which a range of prior distributions, reflecting different assumptions about the genetic architecture of the trait, is assigned to the marker effects (Meuwissen *et al.* 2001; Park and Casella 2008; Kizilkaya *et al.* 2010; Habier *et al.* 2011; Erbe *et al.* 2012; Moser *et al.* 2015b; Gianola and Fernando 2019). For example, it is sometimes desirable to model the majority of marker effects as being zero, and to allow occasional markers with large effects. For some traits, such priors are more biologically meaningful than SNP-BLUP and have been widely used in genomic prediction and GWAS.

In this paper, we demonstrate the effectiveness of HEGP using both SNP-BLUP and BayesC$\pi$ (Kizilkaya *et al.* 2010; Habier *et al.*

2011). BayesC$\pi$ is a representative of the other "Bayesian Alphabet" models, so the extension of HEGP to other priors for marker effects (Meuwissen *et al.* 2001; Park and Casella 2008; Erbe *et al.* 2012; Moser *et al.* 2015b; Gianola and Fernando 2019) does not present further challenges. BayesC$\pi$ (Kizilkaya *et al.* 2010; Habier *et al.* 2011) is typical in that it assigns mixture priors to marker effects, which are multiplied by the Gaussian likelihood of the data to generate the posterior. The BayesC$\pi$ model must be fitted using Gibbs sampling, and therefore we must show that HEGP does not perturb the algorithm and produces numerically stable and accurate estimates.

At each step of the Gibbs sampler, a given unknown is sampled from its full conditional posterior distributions given the latest sampled values of all other unknowns. Below we will show that the full conditional posterior distributions of marker effects are identical when using raw or encrypted data, such that the same posterior distributions will be obtained (this also holds for other parameters of interest). Derivations for other parameters of interest in SNP-BLUP and BayesC$\pi$ can be found in the Appendix.

## Unchanged likelihood using HEGP

In HEGP, the plaintext phenotypes, covariates, and genotype dosages, and the design matrix for fixed effects are encrypted by premultiplication by the same random orthogonal matrix, $\mathbf{P}$. The mixed model using cyphertext for both SNP-BLUP and BayesC$\pi$ can be written as

$$\mathbf{y}^* = \mathbf{X}^*\boldsymbol{\beta} + \mathbf{M}^*\boldsymbol{\alpha} + \mathbf{e}^*. \tag{7}$$

In this model, $\mathbf{y}^* = \mathbf{Py}$ are the encrypted phenotypes, $\mathbf{M}^* = \mathbf{PM} = \mathbf{u}^*$ are the encrypted genotypes, and $\mathbf{X}^* = \mathbf{PX}$ is the encrypted design matrix for fixed effects. After encryption, the residual variance remains unchanged, represented as $\mathbf{var}(\mathbf{e}^*) = \mathbf{var}(\mathbf{Pe}) = \mathbf{P}^T\mathbf{I}\sigma_e^2\mathbf{P} = \mathbf{I}\sigma_e^2$.

The genetic variance becomes $\mathbf{var}(\mathbf{u}^*) = \mathbf{P}^T\mathbf{G}\mathbf{P}\sigma_u^2$ after encryption.

We next show that the likelihood of the data is invariant under orthogonal transformation. Define the plaintext variance matrix $\mathbf{V} = \mathbf{G}\sigma_u^2 + \mathbf{I}\sigma_e^2$ and its cyphertext equivalent $\mathbf{V}^* = \mathbf{P}^T(\mathbf{G}\sigma_u^2 + \mathbf{I}\sigma_e^2)\mathbf{P}$. Then the determinant of the variance matrix is invariant because $|\mathbf{V}^*| = |\mathbf{P}^T\mathbf{V}\mathbf{P}| = |\mathbf{P}^T||\mathbf{V}||\mathbf{P}| = |\mathbf{V}|$ and hence the Gaussian log-likelihood of the plaintext ($\log L$) equals that of the cyphertext ($\log L^*$):

$$
\begin{aligned}
-2\log L(\beta) &= (\mathbf{y} - \mathbf{X}\beta)^T\mathbf{V}^{-1}(\mathbf{y} - \mathbf{X}\beta) + n\log(|\mathbf{V}|) \\
&= (\mathbf{y} - \mathbf{X}\beta)^T(\mathbf{P}^T\mathbf{P})\mathbf{V}^{-1}(\mathbf{P}^T\mathbf{P})(\mathbf{y} - \mathbf{X}\beta) + n\log(|\mathbf{V}|) \\
&= (\mathbf{P}(\mathbf{y} - \mathbf{X}\beta))^T(\mathbf{P}^T\mathbf{V}\mathbf{P})^{-1}(\mathbf{P}(\mathbf{y} - \mathbf{X}\beta)) + n\log(|\mathbf{V}^*|) \\
&= (\mathbf{y}^* - \mathbf{X}^*\beta)^T\mathbf{V}^{*-1}(\mathbf{y}^* - \mathbf{X}^*\beta) + n\log(|\mathbf{V}^*|) \\
&= -2\log L^*(\beta).
\end{aligned}
\tag{8}
$$

Hence, all parameter inference in SNP-BLUP is invariant under orthogonal transformation in the mixed model, resulting in unchanged estimates for $\beta$, for the variance components $\sigma_e^2$, $\sigma_u^2$ and heritability $h^2$.

## Inference of unknowns in mixed model

In BayesC$\pi$, the prior for the marker effects is a mixture of a point mass at zero and a univariate normal distribution with a null mean and a common locus variance $\sigma_\alpha^2$. The full conditional posterior distribution of the marker effect for locus $j$ when it is nonzero (i.e. the full conditional posterior distribution of the marker effect for locus $j$ in SNP-BLUP) can be expressed as

$$
\left(\alpha_j \mid \text{ELSE}\right) \sim N\left(\hat{\alpha}_j, \frac{\sigma_e^2}{\mathbf{m}_j^T\mathbf{m}_j + \frac{\sigma_e^2}{\sigma_\alpha^2}}\right),
\tag{9}
$$

where ELSE stands for all the other parameters and $\hat{\alpha}_j$ is the solution to

$$
\left(\mathbf{m}_j^T\mathbf{m}_j + \frac{\sigma_e^2}{\sigma_\alpha^2}\right)\hat{\alpha}_j = \mathbf{m}_j^T\left(\mathbf{y} - \mathbf{X}\beta - \sum_{j' \neq j}\mathbf{m}_{j'}\alpha_{j'}\right).
\tag{10}
$$

When encrypted genotypic and phenotypic data are used, the full conditional posterior distribution of $\alpha_j$, when it is nonzero, can be written as

$$
\left(\alpha_j \mid \text{ELSE}\right) \sim N\left(\hat{\alpha}_j^*, \frac{\sigma_e^2}{(\mathbf{m}_j^*)^T(\mathbf{m}_j^*) + \frac{\sigma_e^2}{\sigma_\alpha^2}}\right),
\tag{11}
$$

where ELSE stands for all the other parameters, and $\hat{\alpha}_j^*$ is the solution to

$$
\begin{aligned}
\left((\mathbf{m}_j^*)^T(\mathbf{m}_j^*) + \frac{\sigma_e^2}{\sigma_\alpha^2}\right)\hat{\alpha}_j^* &= (\mathbf{m}_j^*)^T\left(\mathbf{y}^* - \mathbf{X}^*\beta - \sum_{j' \neq j}\mathbf{m}_{j'}^*\alpha_{j'}\right) \\
&= (\mathbf{m}_j^*)^T\mathbf{y}^* - (\mathbf{m}_j^*)^T\mathbf{X}^*\beta - \sum_{j' \neq j}(\mathbf{m}_j^*)^T\mathbf{m}_{j'}^*\alpha_{j'}.
\end{aligned}
\tag{12}
$$

We have previously shown that $(\mathbf{m}_j^*)^T(\mathbf{m}_k^*) = \mathbf{m}_j^T\mathbf{m}_k$. Similarly,

$$
\begin{aligned}
(\mathbf{m}_j^*)^T\mathbf{y}^* &= (\mathbf{P}\mathbf{m}_j)^T\mathbf{P}\mathbf{y} \\
&= \mathbf{m}_j^T\mathbf{y}
\end{aligned}
\tag{13}
$$

and

$$
\begin{aligned}
(\mathbf{m}_j^*)^T\mathbf{X}^* &= (\mathbf{P}\mathbf{m}_j)^T\mathbf{P}\mathbf{X} \\
&= \mathbf{m}_j^T\mathbf{X}.
\end{aligned}
\tag{14}
$$

Therefore, the full conditional posterior distribution of $\alpha_j$ using cyphertext, as per equations (11) and (12), is identical to that obtained using the plaintext, as shown in equations (9) and (10). Thus, because HEGP does not change the full conditional posterior distributions in Gibbs sampling, the posterior distributions of marker effects are also identical using plaintext or cyphertext. The same conclusion holds for all other parameters of interest (see Appendix). Note that once estimates of marker effects are obtained, the plaintext of genotypes, if available, should be used to calculate the EBV.

## Joint analysis using encrypted data from multiple contributors

A single research study may only contain a limited amount of data that is underpowered for genetic analysis. This issue can be mitigated through joint analyses using data from multiple studies, e.g. Yengo et al. (2022). An attractive feature of HEGP is that it allows each component of the joint data to be encrypted independently. Thus, each contributor generates its own private key and uses it to encrypt its own plaintext prior to sharing it for joint analysis. The keys are never shared. The process for joint analysis then proceeds as described in the following.

For clarity, let us assume there are three contributors. Assuming variance components, such as the marker effect variance and the residual variance, are identical for all parties, the mixed model for the joint analysis of cyphertext can be written as

$$
\begin{bmatrix} \mathbf{P}_1\mathbf{y}_1 \\ \mathbf{P}_2\mathbf{y}_2 \\ \mathbf{P}_3\mathbf{y}_3 \end{bmatrix} = \begin{bmatrix} \mathbf{P}_1\mathbf{X}_1 \\ \mathbf{P}_2\mathbf{X}_2 \\ \mathbf{P}_3\mathbf{X}_3 \end{bmatrix}\beta + \begin{bmatrix} \mathbf{P}_1\mathbf{M}_1 \\ \mathbf{P}_2\mathbf{M}_2 \\ \mathbf{P}_3\mathbf{M}_3 \end{bmatrix}\alpha + \begin{bmatrix} \mathbf{P}_1\mathbf{e}_1 \\ \mathbf{P}_2\mathbf{e}_2 \\ \mathbf{P}_3\mathbf{e}_3 \end{bmatrix},
\tag{15}
$$

where the matrices related to the $t$th contributor are labeled with subscript "$t$". This equation can be rewritten as

$$
\mathbf{P}\begin{bmatrix} \mathbf{y}_1 \\ \mathbf{y}_2 \\ \mathbf{y}_3 \end{bmatrix} = \mathbf{P}\begin{bmatrix} \mathbf{X}_1 \\ \mathbf{X}_2 \\ \mathbf{X}_3 \end{bmatrix}\beta + \mathbf{P}\begin{bmatrix} \mathbf{M}_1 \\ \mathbf{M}_2 \\ \mathbf{M}_3 \end{bmatrix}\alpha + \mathbf{P}\begin{bmatrix} \mathbf{e}_1 \\ \mathbf{e}_2 \\ \mathbf{e}_3 \end{bmatrix},
\tag{16}
$$

where $\mathbf{P}$ is a block-diagonal orthogonal matrix

$$
\mathbf{P} = \begin{bmatrix} \mathbf{P}_1 & \mathbf{0} & \mathbf{0} \\ \mathbf{0} & \mathbf{P}_2 & \mathbf{0} \\ \mathbf{0} & \mathbf{0} & \mathbf{P}_3 \end{bmatrix}.
\tag{17}
$$

Thus, conceptually, the stacked cyphertexts are equivalent to the stacked plaintexts after encryption by the block-diagonal matrix $\mathbf{P}$, which is the orthogonal matrix assembled from the component random orthogonal matrices $\mathbf{P}_1$, $\mathbf{P}_2$, and $\mathbf{P}_3$. Thus, as shown in the previous section, unknowns inferred from joint cyphertext will be identical to those inferred using the joint

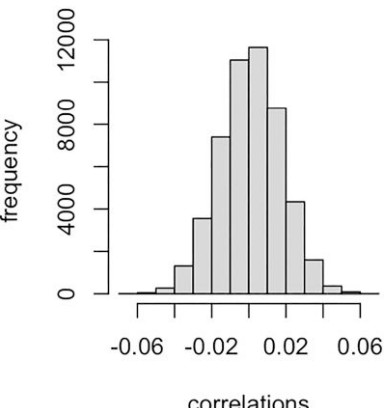

**Fig. 4.** Distribution of correlations between 50,436 pairs of plaintext and cyphertext genotypes using pig dataset in Cleveland *et al.* (2012) ($n = 3534$, $p = 50, 436$). The average correlation is less than 0.001.

plaintext. Note that genomic predictions of the original individuals require the plaintext genotypes, and thus, each contributor can only generate these for their own individuals, but using estimates of marker effects obtained from the combined data for additional accuracy of predictions.

## Security of HEGP

The correlation between the centered plaintext genotypes, represented as $\mathbf{m}_j$, and the cyphertext genotypes, represented as $\mathbf{Pm}_j$, is proportional to $\mathbf{m}_j^T \mathbf{Pm}_j$. When $\mathbf{P}$ is "far from" an identity matrix (or scaled identity matrix), such correlations resemble those between two random vectors. For example, using the pig genotypes from Cleveland *et al.* (2012) ($n = 3, 534$, $p = 50, 436$), the empirical distribution of Pearson correlations between the plaintext and cyphertext genotypes for each marker is shown in Fig. 4. On average, the correlation between raw and encrypted genotypes is about 0.001, which is very close to 0, and almost all ($\sim 93\%$) correlations are inside the interval $[-0.03, 0.03]$. Thus, without decryption, cyphertext genotypes and phenotypes are uninterpretable.

### Decryption without knowledge of the key

To obtain the raw genotypes from the encrypted genotypes, an orthogonal matrix $\mathbf{Q} \sim \mathbf{P}^T$ should be estimated as the key for decryption. When $\mathbf{Q} = \mathbf{P}^T$, exactly, the plaintext genotypes will be recovered, since then $\mathbf{QM}^* = \mathbf{P}^T \mathbf{PM} = \mathbf{M}$. The distance between $\mathbf{QM}^*$ and $\mathbf{M}$ measures how close the decryption is to the raw data. However, because neither $\mathbf{M}$ nor $\mathbf{P}$ are shared, it is difficult to evaluate attempted decryption without a suitable objective function to minimize. Assuming the distance between the attempted decrypted genotype matrix and the plaintext genotype matrix is known (although it is unknown in practice), several strategies to decrypt the genotypes were discussed in Mott *et al.* (2020b). First, in a brute-force approach, numerous random orthogonal matrices (i.e. keys) were generated for decryption. However, massive computing resources would be required to generate and test all possible keys. Mott *et al.* (2020b) reported that even for a dataset with 8 individuals, they could not brute-force the key. A second approach to uncover the decryption key relies on the trimodal distribution of the plaintext genotype frequencies of each marker, assuming all markers are in Hardy–Weinberg equilibrium with publicly available allele frequencies (such as Fig. 2c). Mott *et al.* (2020b) attempted to infer the key by maximizing the kernel density estimator of those non-Gaussian

distributions. However, the results were unsuccessful. Finally, a decryption challenge for HEGP (Mott *et al.* 2020), in which attempts were invited to decrypt HEGP-encrypted plaintext genotypes, has so far failed to elicit a successful attack. More discussion can be found in Mott *et al.* (2020b).

The only identified weakness of HEGP occurs when the data includes variants that are private to an individual. In an extreme case, when each individual has a private variant coded as 1, the plaintext genotype matrix can be written as $\mathbf{M} = [\mathbf{I} \mid \mathbf{M}_{sub}]$, where $\mathbf{I}$ represents genotypes of the $n$ private variants, and $\mathbf{M}_{sub}$ represents genotypes of all the other markers. In this situation, $\mathbf{P}$ itself will be included in the encrypted genotypes since $\mathbf{M}^* = [\mathbf{P} \mid \mathbf{M}_{sub}^*]$. In practice, this extreme case can be avoided by using common variants. However, it suggests that useful information might be extracted from the encrypted data of lower frequency variants, suggesting it is best to remove any variant with a frequency under 0.01 or that is private to fewer than about 10 individuals. Since these variants are typically removed during quality control processing, there should be minimal loss of information.

## Data analysis

The pig dataset in Cleveland *et al.* (2012) was used to validate the equivalent outcomes from both genomic prediction and GWAS analyses using plaintext and cyphertext. This dataset contains 3534 genotyped individuals and the number of SNP markers is 50,436. We simulated phenotypes based on different values for heritability and numbers of quantitative trait loci (QTL) (i.e. causal variants). In detail, phenotypes with heritability equal 0.1, 0.3, 0.5, and 0.7 were simulated, and 1, 10, 50, and 100% of SNPs were randomly sampled as QTL (16 scenarios). Contemporary group effects were included to simulate phenotypes on individuals from 4 groups. For each simulated scenario, 10 replicates were applied. The genotypes of each marker were centered to have 0 mean. The incidence matrix of fixed effects, the genotypes, and the simulated phenotypes were encrypted using a random orthogonal matrix generated as described above. SNP-BLUP and BayesC$\pi$ were applied to analyze the plaintext and cyphertext using the JWAS package (Cheng *et al.* 2018, 2022). In all scenarios, 500,000 Markov chain Monte Carlo (MCMC) iterations were applied to ensure convergence.

We first show that the estimated marker effects ($\hat{\boldsymbol{\alpha}}$) remain unchanged with the cyphertext. Using the plaintext of genotypes ($\mathbf{M}$), the EBV are calculated as $\mathbf{M}\hat{\boldsymbol{\alpha}}$, confirming that the EBVs also remain unchanged. Below we only present the results from BayesC$\pi$, and the conclusions drawn were consistent with those from RR-BLUP results.

## Results
### Estimated marker effects and breeding values

Overall, the marker effects estimated from plaintext and cyphertext were very similar, with a Pearson correlation of 0.9929. The results from 1 replicate in the scenario with $h^2 = 0.3$, QTL%=1% are presented in Fig. 5a, and similar results were observed across all other scenarios. The results for all scenarios are detailed in Table 1, where each value represents the averaged correlation across 10 replicates.

The EBV for all individuals with genotypes $\mathbf{M}$ were calculated as $\mathbf{M}\hat{\boldsymbol{\alpha}}_{plaintext}$, using marker effects estimated from plaintext, and as $\mathbf{M}\hat{\boldsymbol{\alpha}}_{cyphertext}$, using marker effects estimated from cyphertext. Overall, the correlation between EBV calculated using the plaintext and those calculated using the cyphertext was about 0.9996. The results of one replicate from the scenario with

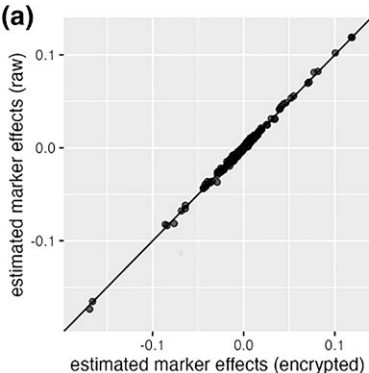
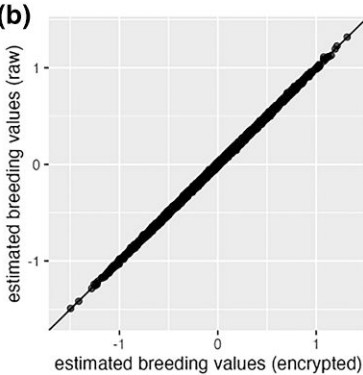
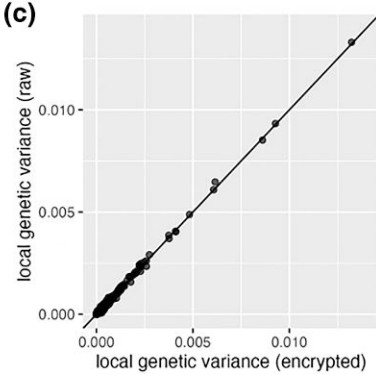

**Fig. 5.** Results from one replicate in the simulation scenario with $h^2 = 0.3$ and QTL%=1%. Each dot represents a pair of results calculated from plaintext (x-axis) vs cyphertext (y-axis). The diagonal line indicates when the plaintext and cyphertext result in the same estimates. a) Comparison between estimated marker effects $\hat{a}_{\text{plaintext}}$ and $\hat{a}_{\text{cyphertext}}$ (correlation = 0.9971). b) Comparison between EBV $\mathbf{M}\hat{a}_{\text{plaintext}}$ and $\mathbf{M}\hat{a}_{\text{cyphertext}}$ (correlation = 0.9997). The pig genotypes data were used ($\mathbf{M}$). c) Comparison between local genetic variances of 2,522 nonoverlapped genomic windows (correlation = 0.9983).

**Table 1.** Pearson correlations between estimated marker effects from plaintext vs ciphertext in different simulation scenarios.

| | | $h^2$ | | | |
| --- | --- | --- | --- | --- | --- |
| | | 0.1 | 0.3 | 0.5 | 0.7 |
| QTL% | 1% | 0.9937 | 0.9950 | 0.9957 | 0.9961 |
| | 10% | 0.9928 | 0.9927 | 0.9926 | 0.9928 |
| | 50% | 0.9898 | 0.9937 | 0.9916 | 0.9911 |
| | 100% | 0.9906 | 0.9932 | 0.9925 | 0.9920 |

Each value is the averaged correlation from 10 replicates.

**Table 2.** Pearson correlations between EBV calculated using marker effects estimated from cyphertext ($\mathbf{M}\hat{a}_{\text{cyphertext}}$) and EBV calculated using the marker effects estimated from plaintext ($\mathbf{M}\hat{a}_{\text{plaintext}}$).

| | | $h^2$ | | | |
| --- | --- | --- | --- | --- | --- |
| | | 0.1 | 0.3 | 0.5 | 0.7 |
| QTL% | 1% | 0.9993 | 0.9997 | 0.9998 | 0.9998 |
| | 10% | 0.9992 | 0.9996 | 0.9997 | 0.9997 |
| | 50% | 0.9992 | 0.9996 | 0.9997 | 0.9997 |
| | 100% | 0.9992 | 0.9995 | 0.9997 | 0.9997 |

Each value represents the average correlation across 10 replicates.

$h^2 = 0.3$, QTL%=1% are shown in Fig. 5b, and similar results were observed for all the other scenarios. The results for all scenarios are listed in Table 2, where each value is the average correlation from 10 replicates.

## Local genetic variances

For GWAS, the genetic variance captured by a genomic window is of interest due to the fact that highly correlated SNPs within a genomic window jointly affect the phenotype, and it is difficult to identify the effect of a single marker (Hayes *et al.* 2010). In GWAS, local genetic variances can be used to estimate window-based posterior probabilities of association (WPPA) (Fernando *et al.* 2017). Here, we divided the pig reference genome into 2,522 nonoverlapping genomic windows, where each window contains about 20 SNPs. The genetic values that are attributed to each

**Table 3.** Pearson correlations between local genetic variances of 2,522 nonoverlapping genomic windows calculated from plaintext or cyphertext.

| | | $h^2$ | | | |
| --- | --- | --- | --- | --- | --- |
| | | 0.1 | 0.3 | 0.5 | 0.7 |
| QTL% | 1% | 0.9954 | 0.9950 | 0.9966 | 0.9968 |
| | 10% | 0.9895 | 0.9923 | 0.9929 | 0.9943 |
| | 50% | 0.9908 | 0.9946 | 0.9932 | 0.9924 |
| | 100% | 0.9728 | 0.9934 | 0.9936 | 0.9933 |

Each value represents the average correlation from 10 replicates.

genomic window were sampled from their posterior distributions using MCMC.

Overall, the correlation between local genetic variances estimated using plaintext or cyphertext was about 0.9923. The results of one replicate from the scenario with $h^2 = 0.3$, QTL%=1% are presented in Fig. 5c, and similar results were observed for all other scenarios. The results for each scenario are listed in Table 3, where each value represents the average correlation from 10 replicates.

## Joint analysis of cyphertext from multiple contributors

To perform joint cyphertext analysis, the 3,534 individuals in the pig dataset were split into 2 datasets ($n_1 = 500$, $n_2 = 3,034$), modeling the scenario of 2 data contributors. The genotypes were independently centered within each contributor to have 0 means. The simulated phenotypes data in the scenario with heritability of 0.3 and 10% QTLs were used. The plaintext phenotypes, genotypes, and covariates were independently encrypted by each contributor. For example, for contributor 1, the encrypted genotype data are $\mathbf{M}_1^* = \mathbf{P}_1\mathbf{M}_1$ with $\mathbf{P}_1$ of size $n_1 \times n_1$, and for contributor 2, the encrypted genotype data are $\mathbf{M}_2^* = \mathbf{P}_2\mathbf{M}_2$ with $\mathbf{P}_2$ of size $n_2 \times n_2$. Only the cyphertexts were shared, not the encryption keys $\mathbf{P}_1$ or $\mathbf{P}_2$. We repeated the previous analyses using the joint cyphertexts and the joint plaintexts. Using the joint cyphertexts yielded results very similar to those using the joint plaintexts.

Moreover, using joint cyphertexts to estimate parameters resulted in significantly higher prediction accuracies than only using the data from a single-data contributor. This is to be expected, as a larger sample size improves parameter inference. For the 500 individuals in contributor 1, we calculated their EBV

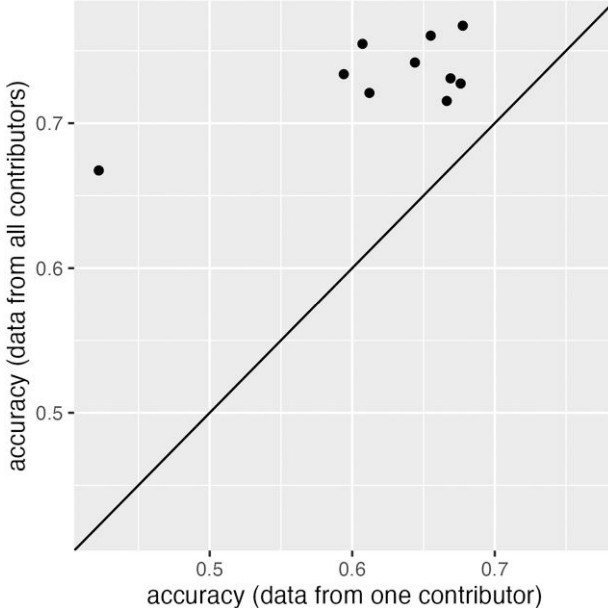

**Fig. 6.** The prediction accuracies of EBV for 500 individuals in a single data contributor. Each dot represents a pair of results calculated either using only this contributor's data (x-axis) or using the joint data from all contributors (y-axis). Joint analyses had significantly higher accuracies than those using data from a single contributor (pairwise *t*-test *P*–value < 0.0005).

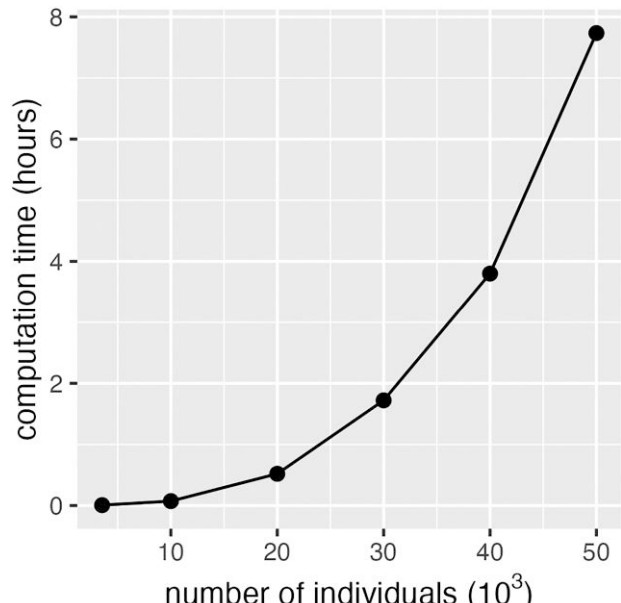

**Fig. 7.** Time to generate a random orthogonal matrix from the Stiefel manifold. The *x*-axis is the size of **P** matrix (i.e. number of individuals) and the *y*-axis is the computation time.

using marker effects estimated from the joint data ($\mathbf{M}_1\hat{\boldsymbol{\alpha}}_{\text{joint}}$), as well as using marker effects estimated from only contributor 1's data ($\mathbf{M}_1\hat{\boldsymbol{\alpha}}_{\text{sub1}}$). The comparison between the accuracy of $\mathbf{M}_1\hat{\boldsymbol{\alpha}}_{\text{joint}}$ and $\mathbf{M}_1\hat{\boldsymbol{\alpha}}_{\text{sub1}}$ is shown in Fig. 6, where each dot represents the result of one replicate. The joint data resulted in significantly higher accuracy than using data from a single contributor (pairwise *t*-test *P*–value < 0.0005). The same conclusions were drawn for contributor 2.

## Discussion
### Overview
In this study, we have built on the HEGP methodology introduced in Mott *et al.* (2020b) to show how it can be extended to the wide class of mixed models including Bayesian Alphabet models that are commonly used in animal and crop quantitative genetics. We have also shown how joint analysis of multiple datasets fits into this framework and confirmed the increase in prediction accuracy of breeding values expected from joint analyses, with plaintext analyses also holds for cyphertext. Note that the estimated marker effects using encrypted data are the same as those using the raw data.

HEGP enables adherence to, and capitalizes on, the benefits of the FAIR principles in genome-to-phenome studies. In the context of animal and crop breeding, it also addresses many of the privacy, intellectual property, and commercial interest issues that prevent the sharing and reuse of data for both research and industry applications.

### Alternatives
The principal alternative strategy to sharing genotype and phenotype data is to share GWAS summary statistics, for example, marker regression coefficient estimates and their standard errors. This approach is most suited to single-marker regression analyses

that are typical in human studies (MacArthur *et al.* 2021). To this end, databases have been built to collect GWAS summary statistics (Welter *et al.* 2014; MacArthur *et al.* 2017; Buniello *et al.* 2019), and methods have been proposed to facilitate large meta-analyses needed for the increased power in dissecting the genetic basis of complex traits (Yang *et al.* 2012; Vilhjálmsson *et al.* 2015; Barbeira *et al.* 2018; Lloyd-Jones *et al.* 2019; Privé *et al.* 2020; Werme *et al.* 2022; Yengo *et al.* 2022). However, meta-analyses based on these summary statistics rely heavily on approximations due to unavailability of the individual-level data. HE methods such as HEGP do not require such approximations, provided they factorize into a prior distribution of the markers multiplied by the Gaussian likelihood of the data, and can be fitted by MCMC methods such as Gibbs sampling.

### Rounding errors
In HEGP, the individuals' plaintext identities, phenotypes, and genotypes are obscured by premultiplying by a high-dimensional random orthogonal matrix. In the resulting cyphertext, the relationships between SNPs, and between SNPs and phenotypes are preserved, but the relationships between individuals are scrambled—in fact, the concept of an individual is nonsensical after encryption, as records in the encrypted data are random linear combinations of the original individuals' records. Theoretically, plaintext and cyphertext should yield identical estimates of marker effects and other parameters, but due to rounding errors, as well as Monte Carlo errors in the case of models using MCMC, the estimated marker effects are not identical but rather very similar, with correlations close to 1.0. In detail, rounding errors occur because the off-diagonal elements of $\mathbf{P}^{\mathsf{T}}\mathbf{P}$ are very small values, $\sim 10^{-13}$. Mott *et al.* (2020b) reported that rounding errors were negligible for **P** with dimensions up to $10,000 \times 10,000$. To alleviate the problem of rounding errors for a very large dataset, **P** can be constructed as a block-diagonal matrix, where each block is a random orthogonal matrix.

## Time complexity

The time taken to generate a random $n \times n$ orthogonal matrix $\mathbf{P}$ from the Stiefel manifold is proportional to $n^3$, where $n$ is the number of individuals, being dominated by the eigen decomposition. The time taken to multiply the plaintext by $\mathbf{P}$ to produce the cyphertext is proportional to $pn^2$, where $p$ is the number of markers. In a computer server with 5 cores, generating $\mathbf{P}$ for the pig dataset ($n = 3,534$) took less than 1 min. For a dataset with 10,000 individuals, the time to generate $\mathbf{P}$ was about 5 min. However, the time to generate $\mathbf{P}$ for 50,000 individuals was about 8 h. As shown in Fig. 7, running time increased rapidly as sample size increased.

In practice, with hundreds of thousands of individuals, many relatively small random orthogonal matrices (e.g. $50,000 \times 50,000$) could be generated in parallel, and then a large block-diagonal orthogonal matrix could be constructed, with each block being a random orthogonal matrix (i.e. a block-diagonal random orthogonal matrix). This larger block-diagonal orthogonal matrix, as well as any permutation of such a matrix, can be used as the encryption key.

The size of the cyphertext is the same as the plaintext and, therefore, the computational effort required for each iteration in MCMC is comparable. In our analysis of the pig dataset (Cleveland *et al.* 2012), the number of MCMC iterations necessary to ensure the convergence of the MCMC process was also similar between analysis of the plaintext and the cyphertext.

## Security

With an appropriately sampled HEGP encryption key, the correlation between raw and encrypted data resembles that between two random vectors. For the pig dataset (Cleveland *et al.* 2012), the absolute Pearson correlation between raw and encrypted marker genotypes was almost always less than 0.03. To increase the security of the encrypting and lower the risk of discovery of the decryption key, genotypes for SNPs with very low minor allele frequencies should not be shared. Since only cyphertext are shared, the unknown nature of the plaintext genotypes $\mathbf{M}$ makes it difficult to evaluate decryption attempts. To date, decryption attacks have been proven ineffective (Mott *et al.* 2020b), even when $\mathbf{M}$ was available for evaluation. However, further exploration is still needed to determine whether HEGP is cryptographically secure.

## Protocols for data sharing in HEGP

Finally, we mention some points to consider when sharing HEGP cyphertext. First, it is necessary for all contributors to agree on a common set of markers and covariates to be shared, and on whether phenotypes are to be residualized by removing covariate effects before sharing—in which case the cyphertext versions of covariates need not be shared—or afterwards, during the joint analysis. Second, missing genotypes, covariates, and phenotypes must be imputed, either for markers not genotyped in a particular contributor's data, or to fill in sporadic missing values (HEGP does not allow missing data). Third, the sharing topology must be agreed upon: each cyphertext could be shared with all contributors, so that each participant could conduct their own analysis, or instead, it could be shared only with a trusted third party who would perform the agreed-upon analysis. Fourth, although the joint analysis' parameter estimates do not need decrypting, the parties may want to agree beforehand on their subsequent use and dissemination.

These considerations would likely require the contributors to set up a protocol for data sharing, reflecting the sensitivity and value of the component datasets, and which will likely vary depending on circumstances and commercial considerations.

Notwithstanding the HEGP-specific technical requirements, such a protocol should be simpler to implement than agreements involving the sharing of plaintext data.

## Data availability

Pig genotypes used in the analysis are publicly available in Cleveland *et al.* (2012). The simulated phenotypes and all scripts are available at https://github.com/zhaotianjing/encryption. The authors state that all data necessary for confirming the conclusions presented in the article are represented fully within the article.

## Funding

This work was supported by a UKRI BBSRC grant BB/V00767X/1 awarded to RM, by Agricultural Genome to Phenome Initiative (AG2PI) under USDA-NIFA awards 2020-70412-32615 and 2021-70412-35233, and by USDA-NIFA awards 2021-67015-33412, 2023-67015-39564, and 2023-70412-41054.

## Conflicts of interest

The author(s) declare no conflicts of interest.

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

*Editor: M. Goddard*

# Appendix

## Homomorphic encryption for single-marker analysis in GWAS

Mott et al. (2020b) proposed the HEGP method for single-marker regression in GWAS using linear mixed models with Gaussian errors, where the raw genotypes and phenotypes data were premultiplied by a high-dimensional random orthogonal matrix. Mott et al. (2020b) showed that such encryption does not change the likelihood of the quantitative trait in the GWAS model, a point we will illustrate further below. For more details, refer to Mott et al. (2020b).

In detail, for $n$ individuals genotyped with $p$ markers, the raw genotypes and phenotypes matrix were premultiplied by the same random orthogonal matrix as

$$\mathbf{M}^* = \mathbf{P}\mathbf{M}$$
$$\mathbf{y}^* = \mathbf{P}\mathbf{y}, \tag{A1}$$

where $\mathbf{M}$ is an $n \times p$ genotype matrix, $\mathbf{y}$ is the vector of phenotypes of length $n$, $\mathbf{P}$ is an $n \times n$ random orthogonal matrix whose columns and rows are orthonormal vectors (i.e. $\mathbf{P}^T = \mathbf{P}^{-1}$), and $\mathbf{M}^*$ and $\mathbf{y}^*$ are encrypted genotypes and phenotypes, respectively. The covariate matrix $\mathbf{X}$ should be encrypted as well, producing $\mathbf{X}^* = \mathbf{P}\mathbf{X}$.

In GWAS, the linear model used to test the significance of $j$th SNP ($j = 1, \ldots, p$) is

$$\mathbf{y} = \mathbf{X}\boldsymbol{\beta} + \mathbf{m}_j\alpha_j + \mathbf{u} + \mathbf{e}$$
$$= \mathbf{X}\boldsymbol{\beta} + \mathbf{m}_j\alpha_j + \boldsymbol{\epsilon}, \tag{A2}$$

where $\mathbf{y}$ is the phenotype, $\mathbf{X}$ is the covariate matrix, $\boldsymbol{\beta}$ is the fixed effects of covariates, $\mathbf{m}_j$ is the (centered and scaled) genotypes of $j$th SNP, $\alpha_j$ is the regression coefficient of $j$th SNP. $\mathbf{u}$ is a random vector for polygenic effects with $\mathbf{u} \sim N(\mathbf{0}, \mathbf{G}\sigma_u^2)$, and $\mathbf{e}$ is a random vector of residuals with $\mathbf{e} \sim N(\mathbf{0}, \mathbf{I}\sigma_e^2)$. Thus, the variance of $\mathbf{y}$ is $\text{var}(\mathbf{y}) = \mathbf{G}\sigma_u^2 + \mathbf{I}\sigma_e^2 = \mathbf{V}$.

Applying orthogonal encryption, the above GWAS model becomes

$$\mathbf{P}\mathbf{y} = \mathbf{P}\mathbf{X}\boldsymbol{\beta} + \mathbf{P}\mathbf{m}_j\alpha_j + \mathbf{P}\boldsymbol{\epsilon}$$
$$\mathbf{y}^* = \mathbf{X}^*\boldsymbol{\beta} + \mathbf{m}_j^*\alpha_j + \boldsymbol{\epsilon}^*, \tag{A3}$$

where the variance of encrypted phenotypes becomes $\text{var}(\mathbf{P}\mathbf{y}) = \mathbf{P}\mathbf{V}\mathbf{P}^T$. We showed in equation (8) that the likelihood is invariant under orthogonal transformation. Thus, HEGP leaves likelihood-based inferences for GWAS model—used to test the significance of a single marker—unaffected. This includes the maximum likelihood parameter estimates and $P$-values for likelihood-based tests of significance.

## Gibbs sampler for the linear mixed model

The full conditional posterior distributions of parameters of interest in SNP-BLUP and BayesC$\pi$ are shown below. More details can be found in Fernando and Garrick (2013).

### Residual variance

The full conditional posterior distribution of residual variance $\sigma_e^2$ follows a scaled inverse chi-square distribution with $n + v_e$ degrees of freedom and scale parameter $\frac{\mathbf{e}^T\mathbf{e} + v_e S_e^2}{n + v_e}$. That is,

$$f(\sigma_e^2 \mid \text{ELSE}) \propto (\sigma_e^2)^{-\frac{n+v_e+2}{2}} \exp\left[-\frac{1}{2\sigma_e^2}(\mathbf{e}^T\mathbf{e} + v_e S_e^2)\right], \tag{A4}$$

where $\mathbf{e}$ is the residuals. Since $(\mathbf{e}^*)^T(\mathbf{e}^*) = \mathbf{e}^T\mathbf{P}^T\mathbf{P}\mathbf{e} = \mathbf{e}^T\mathbf{e}$, the full conditional posterior distribution of $\sigma_e^2$ is unchanged with the encrypted data.

### Marker effect variance

The full conditional posterior distribution of $\sigma_\alpha^2$ follows a scaled inverse chi-square distribution with $k + v_\alpha$ degrees of freedom and scale parameter $\frac{\boldsymbol{\alpha}^T\boldsymbol{\alpha} + v_\alpha S_\alpha^2}{k + v_\alpha}$, where $k = \sum \delta_j$ is the number of markers included in the model. In detail,

$$f(\sigma_\alpha^2 \mid \text{ELSE}) \propto (\sigma_\alpha^2)^{-\frac{k+v_\alpha+2}{2}} \exp\left[-\frac{1}{2\sigma_\alpha^2}(\boldsymbol{\alpha}^T\boldsymbol{\alpha} + v_\alpha S_\alpha^2)\right]. \tag{A5}$$

We have proven that $\boldsymbol{\alpha}$ is unchanged with the encrypted data, and thus, the full conditional posterior distribution of $\sigma_\alpha^2$ is also unchanged.

### Fixed effects

The full conditional posterior distribution of $j$th fixed effects $\beta_j$ follows a univariate normal distribution with mean $\frac{\mathbf{x}_j^T(\mathbf{y} - \mathbf{M}\boldsymbol{\alpha} - \sum_{j' \neq j} \mathbf{x}_{j'}\beta_{j'})}{\mathbf{x}_j^T\mathbf{x}_j}$ and variance $\frac{\sigma_e^2}{\mathbf{x}_j^T\mathbf{x}_j}$.

Given $\mathbf{X}^* = \mathbf{P}\mathbf{X}$, we have

$$[\mathbf{x}_1^*, \ldots, \mathbf{x}_p^*] = \mathbf{P}[\mathbf{x}_1, \ldots, \mathbf{x}_p]$$
$$= [\mathbf{P}\mathbf{x}_1, \ldots, \mathbf{P}\mathbf{x}_p]. \tag{A6}$$

Thus, the $\mathbf{x}_j^T\mathbf{x}_j$ is unchanged using encrypted data since

$$(\mathbf{x}_j^*)^T(\mathbf{x}_j^*) = (\mathbf{P}\mathbf{x}_j)^T(\mathbf{P}\mathbf{x}_j)$$
$$= \mathbf{x}_j^T\mathbf{x}_j. \tag{A7}$$

The $\mathbf{x}_j^T\mathbf{y}$ is also unchanged using encrypted data since

$$(\mathbf{x}_j^*)^T(\mathbf{y}^*) = (\mathbf{P}\mathbf{x}_j)^T\mathbf{P}\mathbf{y}$$
$$= \mathbf{x}_j^T\mathbf{y}. \tag{A8}$$

Similarly, $\mathbf{x}_j^T\mathbf{M}$ and $\mathbf{x}_j^T\mathbf{x}_{j'}$ are also unchanged using encrypted data. Thus, the full conditional posterior distribution of $\beta_j$ is unchanged.

### Indicator variables

In BayesC$\pi$, an indicator Bernoulli variable $\delta_j$ is introduced for locus $j$ that is 1 with probability $1 - \pi$ and 0 with probability $\pi$. The full conditional posterior distribution of indicator variable $\delta_j$ is

$$f(\delta_j = 1 \mid \text{ELSE})$$
$$= \frac{f_1(r_j \mid \sigma_\alpha^2, \sigma_e^2)f(\delta_j = 1)}{f_0(r_j \mid \sigma_e^2)f(\delta_j = 0) + f_1(r_j \mid \sigma_\alpha^2, \sigma_e^2)f(\delta_j = 1)}, \tag{A9}$$

where $f_1(r_j \mid \sigma_\alpha^2, \sigma_e^2)$ is a univariate normal distribution with

$$E(r_j \mid \sigma_\alpha^2, \sigma_e^2) = 0, \text{Var}(r_j \mid \sigma_\alpha^2, \sigma_e^2) = (\mathbf{m}_j^T\mathbf{m}_j)^2\sigma_\alpha^2 + \mathbf{m}_j^T\mathbf{m}_j\sigma_e^2$$

and $f_0(r_j \mid \sigma_e^2)$ is a univariate normal distribution with

$$E(r_j \mid \sigma_e^2) = 0, \text{Var}(r_j \mid \sigma_e^2) = \mathbf{m}_j^T\mathbf{m}_j\sigma_e^2$$

and

$$r_j = \mathbf{m}_j^T \left( \mathbf{y} - \mathbf{X}\boldsymbol{\beta} - \sum_{j' \neq j} \mathbf{m}_{j'} \alpha_{j'} \delta_{j'} \right).$$

We have showed that $r_j$ and $\mathbf{m}_j^T \mathbf{m}_j$ are unchanged with encrypted data. Thus, the full conditional posterior distribution of $\delta_j$ is unchanged.

*Inclusion probabilities*

In BayesC$\pi$, the full conditional posterior distribution of inclusion probability $\pi$ follows a Beta distribution with shape parameter $p - k + 1$ and $k + 1$. That is,

$$f(\pi \mid \text{ELSE}) \propto \pi^{(p-k)}(1 - \pi)^k. \tag{A10}$$

Using encrypted data will not affect the full conditional posterior distribution of $\pi$.