## [Peer Review File · Genetics]

Using encrypted genotypes and phenotypes for collaborative genomic analyses to maintain data confidentiality

Tianjing Zhao, Fangyi Wang, Richard Mott, Jack Dekkers, and Hao Cheng

NOTE: The reviews and decision letters are unedited and appear as submitted by the reviewers.

In extremely rare instances and as determined by a Senior Editor or the EIC, portions of a review may be redacted. If a review is signed, the reviewer has agreed to no longer remain anonymous.

The review history appears in chronological order.

Review Timeline:

Submission Date:	2023-10-09
Editorial Decision:	2023-10-30
Revision Received:	2023-11-07
Accepted:	2023-11-13

October 30, 2023

RE: GENETICS-2023-306542

Dear Dr. Cheng:

I am pleased to accept your manuscript entitled "Using encrypted genotypes and phenotypes for collaborative genomic analyses to maintain data confidentiality" for publication in GENETICS, pending minor revision.

Please submit your revision along with a brief description of how you modified the manuscript in response to the reviewers' concerns and suggestions (which can be viewed at the bottom of this email). I expect you should be able to submit a revised manuscript within 30 days. A suitably revised manuscript will be acceptable for publication; I don't expect to send it out for review.

Please ensure that you have included a Data Availability Statement at the end of the Materials and Methods section. Details available at <https://academic.oup.com/genetics/content/prep-manuscript>. The DAS should include the accession numbers or DOIs of any data you have placed in public repositories, describe supplemental material, include applicable IRB numbers, and may include specifications for how to properly acknowledge or cite the data.

When revising the ms., please make an effort to shorten it, because that almost always improves a manuscript. We urge authors to heed the advice of Strunk and White: "omit needless words"¹. Follow this link to submit the revised manuscript: Link Not Available

Thank you for submitting this story to Genetics.

Sincerely,

Michael Goddard
Associate Editor
GENETICS

Approved by:
Sharon Browning
Senior Editor
GENETICS

Reviewer comments:

Reviewer #1 (Comments for the Authors (Required)):

The manuscript "Using encrypted genotypes and phenotypes for collaborative genomic analyses to maintain data confidentiality" by Zhao et al. shows how an already developed method, HEGP, can be applied to various situations. This method looks promising to improve the reproducibility in biology and genetics field, in particular when concerns about the data are present, such as when using human data or in collaboration between public and private entities.

I found the manuscript clear and well written. My main comment on this manuscript is what does it add to the previous article by Mott et al (2020) that describe the HEGP method. Except if I missed something, this article only shows additional applications of the HEGP method and not an extension of the method.

I've got some additional minor comments:

- l.275: instead of writing "almost all correlations are inside the interval $[-0.03, 0.03]$ ", wouldn't it be more precise to give the 95% confidence interval of the distribution?

- legend of Figure 5: instead of writing "The diagonal line was used for reference such that a dot close to this line represents similar estimates.", wouldn't it be more straightforward to directly write "The diagonal line shows when the plaintext and

cyphertext result in the same estimate", or even " $y = x$ "?

- I know that all the results are available on the author github page (highly appreciated by the way), but I also think that having the corresponding figures directly available as supplementary material could increase the ease of reading.

Reviewer #2 (Comments for the Authors (Required)):

This is an interesting paper that further develops a concept of encrypting genomic and phenotypic data, enabling exchange of such data in an encrypted form, while still being able to do e.g. combined genomic prediction accuracies.

Main comments:

One omission is that the described analyses combining data use univariate models. In other words, they assume that the different datasets have a genetic correlation between them of 1. When combining datasets one may want to challenge such assumption by running e.g. a bivariate model to estimate the genetic correlation between the datasets.

It might be good to add a few sentences in the paper to explicitly indicate some procedures or packages to compute matrix P. I know that readers can derive this from the script that was provided, but it would be good to have an explicit statement in the paper itself.

Some software packages for genomic prediction may take advantage of the usual 0,1,2 nature of SNP genotype data, for instance by efficient bitwise storing of the genotypes. After encryption, genotypes are Gaussian, which implies this is no longer possible, and genotypes need to be stored as real numbers. This can lead to considerably higher RAM usage. It would be good to comment on this in the paper.

In your simulation you used 1, 10, 50 or 100% of the SNPs as QTL. It seems that you left the SNPs sampled as QTL in the analysis. I guess that hasn't affected your results? Please comment on this.

The results described in lines 404-416 are very obvious, as it is well-known that combining data generally leads to larger prediction accuracy.

It is stated in lines 522-523 that "missing genotypes, covariates and phenotypes must be imputed". Does this imply that the method cannot be applied for (ssGBLUP) analyses of datasets where e.g. only part of the individuals with phenotypes also have genotypes? Please mention this in the manuscript.

Specific comments:

Between lines 221 and 222:

- consider to add "(LogL)" after "log-likelihood of the plaintext" and "(LogL*)" after "that of the cyphertext", if that makes sense.
- In the formula, there seems to be a typo. I think " $V^* 1$ " should be " V^*-1 "?

Line 245: "federated analyses" I don't know what that means. This may just be me.

Line 344 "as $\hat{\alpha}$, which equals α " It is not clear what " α " (without the hat) is in this case.

Line 394: delete "a" ?

Line 476: ", " => ", "

Reviewer #3 (Comments for the Authors (Required)):

Authors present a method of encryption suitable for phenotypes and genotypes to facilitate sharing sensitive data among research groups. They demonstrate that the characteristics of interest in the data such as variance components and SNP effect estimates are retained but individual identification is obscured. Proofs of the retention of the data characters are provided as well as a pig-based example. The only issue I found was in the section starting in line 163, there is a "?" among the citations. Perhaps it is a flag for additional checking. The proposed procedure could be of considerable importance in that combined data can lead to more accurate and comprehensive results.

Associate Editor comments:

Associate Editor

I am pleased to accept your manuscript entitled "Using encrypted genotypes and phenotypes for collaborative genomic analyses to maintain data confidentiality" for publication in GENETICS, pending minor revision. Please submit your revision along with a brief description of how you modified the manuscript in response to the reviewers' concerns and suggestions

Response:

We appreciate the valuable feedback from the reviewers and have made an effort to incorporate all the suggested changes and corrections.

Reviewer #1 (Comments for the Authors (Required)):

The manuscript "Using encrypted genotypes and phenotypes for collaborative genomic analyses to maintain data confidentiality" by Zhao et al. shows how an already developed method, HEGP, can be applied to various situations. This method looks promising to improve the reproducibility in biology and genetics field, in particular when concerns about the data are present, such as when using human data or in collaboration between public and private entities.

1. I found the manuscript clear and well-written. My main comment on this manuscript is what does it add to the previous article by Mott et al (2020) that describe the HEGP method. Except if I missed something, this article only shows additional applications of the HEGP method and not an extension of the method.

Response: Thanks for your comment. We have mentioned in our manuscript that Mott et al (2022) proposed HEGP specifically for single-marker regression in GWAS using linear mixed models with Gaussian errors. In this manuscript, we extended the HEGP scheme for wider application to many popular mixed models beyond single-marker regression, including Bayesian variable selection methods (i.e., Bayesian Alphabet) for genetic parameter estimation, genomic prediction and GWAS.

I've got some additional minor comments:

2. I.275: instead of writing "almost all correlations are inside the interval $[-0.03, 0.03]$ ", wouldn't it be more precise to give the 95% confidence interval of the distribution?

Response: thanks for your suggestion. We have added the percentage in this sentence to make it more precise.

3. legend of Figure 5: instead of writing "The diagonal line was used for reference such that a dot close to this line represents similar estimates.", wouldn't it be more straightforward to directly write "The diagonal line shows when the plaintext and cyphertext result in the same estimate", or even " $y = x$ "?

Response: thanks for your suggestion. We have modified this sentence.

4. I know that all the results are available on the author github page (highly appreciated by the way), but I also think that having the corresponding figures directly available as supplementary material could increase the ease of reading.

Response: thanks for your suggestion. We have provided tables to show that results from plaintext and cyphertext were very similar, so the figures for all replicates have the same patterns. Considering there are too many figures (4 heritability * 4 QTL percentage * 10 replicates = 160 figures), we prefer to put one figure in our manuscript to represent all other figures.

Reviewer #2 (Comments for the Authors (Required)):

This is an interesting paper that further develops a concept of encrypting genomic and phenotypic data, enabling exchange of such data in an encrypted form, while still being able to do e.g. combined genomic prediction accuracies.

Main comments:

One omission is that the described analyses combining data use univariate models. In other words, they assume that the different datasets have a genetic correlation between them of 1. When combining datasets one may want to challenge such assumption by running e.g. a bivariate model to estimate the genetic correlation between the datasets.

It might be good to add a few sentences in the paper to explicitly indicate some procedures or packages to compute matrix P. I know that readers can derive this from the script that was provided, but it would be good to have an explicit statement in the paper itself.

Response: thanks for your suggestion. We have added one sentence in line 141: The R package `rstiefel` (Hoff 2012) can be used to generate P.

Some software packages for genomic prediction may take advantage of the usual 0,1,2 nature of SNP genotype data, for instance by efficient bitwise storing of the genotypes.

After encryption, genotypes are Gaussian, which implies this is no longer possible, and genotypes need to be stored as real numbers. This can lead to considerably higher RAM usage. It would be good to comment on this in the paper.

Response: Thanks for your comment. As you mentioned, some software packages may take advantage of the encoding of SNP genotype data (0/1/2) to efficiently store the genotype matrix. For example, the PLINK binary file format stores the genotypes of four individuals in one byte (2 bits per genotype). In HEGP, after encryption, the genotype data follow Gaussian distributions, thus the above storage approach cannot be applied to encrypted data, and leads to higher memory usage. However, centered genotypes are usually used in the statistical algorithm, which requires converting such integers into real numbers. Thus, additional investigation is needed to study this and it is out of the scope of our paper.

In your simulation you used 1, 10, 50 or 100% of the SNPs as QTL. It seems that you left the SNPs sampled as QTL in the analysis. I guess that hasn't affected your results? Please comment on this.

Response: Yes, this will not affect the results.

The results described in lines 404-416 are very obvious, as it is well-known that combining data generally leads to larger prediction accuracy.

Response: Yes, so we only use one example to demonstrate it.

It is stated in lines 522-523 that "missing genotypes, covariates and phenotypes must be imputed". Does this imply that the method cannot be applied for (ssGBLUP) analyses of datasets where e.g. only part of the individuals with phenotypes also have genotypes? Please mention this in the manuscript.

Response: We think HEGP can be applied to the single-step approach, and we will investigate it in future studies.

Specific comments:

Between lines 221 and 222:

- consider to add "(LogL)" after "log-likelihood of the plaintext" and "(LogL*)" after "that of the cyphertext", if that makes sense.

Response: thanks, we have added them.

- In the formula, there seems to be a typo. I think " $V^* 1$ " should be " V^*-1 "?

Response: I have double-checked the formula, and I cannot find the " $V^* 1$ " you mentioned.

Line 245: "federated analyses" I don't know what that means. This may just be me.

Response: thanks, we have changed it to "joint analysis"

Line 344 "as $\hat{\alpha}$, which equals α " It is not clear what " α " (without the hat) is in this case.

Response: thanks, we have modified this sentence.

Line 394: delete "a" ?

Response: thanks, we have deleted the "a".

Line 476: ", ," => ", "

Response: thanks, we have modified it.

Reviewer #3 (Comments for the Authors (Required)):

Authors present a method of encryption suitable for phenotypes and genotypes to facilitate sharing sensitive data among research groups. They demonstrate that the characteristics of interest in the data such as variance components and SNP effect estimates are retained but individual identification is obscured. Proofs of the retention of the data characters are provided as well as a pig-based example.

1. The only issue I found was in the section starting in line 163, there is a "?" among the citations. Perhaps it is a flag for additional checking. The proposed procedure could be of considerable importance in that combined data can lead to more accurate and comprehensive results.

Response: thanks, we have corrected it, and we have checked the manuscript.

November 8, 2023

RE: GENETICS-2023-306542R1

Dr. Hao Cheng
University of California - Davis
Animal Science
1 Shield Avenue
Davis, California 95616

Dear Dr. Cheng:

Congratulations! We are delighted to inform you that your manuscript entitled "Using encrypted genotypes and phenotypes for collaborative genomic analyses to maintain data confidentiality" is acceptable for publication in GENETICS. Many thanks for submitting your research to the journal.

To Proceed to Production:

1. Format your article according to GENETICS style, as discussed at <https://academic.oup.com/genetics/pages/general-instructions>, and upload your final files at <https://genetics.msubmit.net>.
2. Your manuscript will be published as-is (unedited-as submitted, reviewed, and accepted) at the GENETICS website as an Advanced Access article and deposited into PubMed shortly after receipt of source files and the completed license to publish. Please notify sourcefiles@thegsajournals.org if you do not wish to publish your article via Advanced Access.
3. We invite you to submit an original color figure related to your paper for consideration as cover art. Please email your submission to the editorial office or upload it with your final files. You can submit a small-sized image for evaluation, and if selected, the final image must be a TIFF file 2513px wide by 3263px high (8.375 by 10.875 inches; resolution of 600ppi). Please avoid graphs and small type.

If you have any questions or encounter any problems while uploading your accepted manuscript files, please email the editorial office at sourcefiles@thegsajournals.org.

Sincerely,

Michael Goddard
Associate Editor
GENETICS

Approved by:
Sharon Browning
Senior Editor
GENETICS

note: Please add jnls.author.support@oup.com and genetics.oup@kwglobal.com (or the domains @oup.com and @kwglobal.com) to your email program's "safe senders" list. You will be contacted by both at various points during the production process.

Review comments (if applicable):